# Hypertension in People Exposed to Environmental Cadmium: Roles for 20-Hydroxyeicosatetraenoic Acid in the Kidney

**DOI:** 10.3390/jox15040122

**Published:** 2025-08-01

**Authors:** Soisungwan Satarug

**Affiliations:** Centre for Kidney Disease Research, Translational Research Institute, Woolloongabba, Brisbane, QLD 4102, Australia; sj.satarug@yahoo.com.au

**Keywords:** albuminuria, blood pressure, cadmium, glomerular filtration rate, hypertension, 20-hydroxyeicosatetraenoic acid, kidney disease

## Abstract

Chronic kidney disease (CKD) has now reached epidemic proportions in many parts of the world, primarily due to the high incidence of diabetes and hypertension. By 2040, CKD is predicted to be the fifth-leading cause of years of life lost. Developing strategies to prevent CKD and to reduce its progression to kidney failure is thus of great public health significance. Hypertension is known to be both a cause and a consequence of kidney damage and an eminently modifiable risk factor. An increased risk of hypertension, especially among women, has been linked to chronic exposure to the ubiquitous food contaminant cadmium (Cd). The mechanism is unclear but is likely to involve its action on the proximal tubular cells (PTCs) of the kidney, where Cd accumulates. Here, it leads to chronic tubular injury and a sustained drop in the estimated glomerular filtration rate (eGFR), a common sequela of ischemic acute tubular necrosis and acute and chronic tubulointerstitial inflammation, all of which hinder glomerular filtration. The present review discusses exposure levels of Cd that have been associated with an increased risk of hypertension, albuminuria, and eGFR ≤ 60 mL/min/1.73 m^2^ (low eGFR) in environmentally exposed people. It highlights the potential role of 20-hydroxyeicosatetraenoic acid (20-HETE), the second messenger produced in the kidneys, as the contributing factor to gender-differentiated effects of Cd-induced hypertension. Use of GFR loss and albumin excretion in toxicological risk calculation, and derivation of Cd exposure limits, instead of β_2_-microglobulin (β_2_M) excretion at a rate of 300 µg/g creatinine, are recommended.

## 1. Introduction

Exposure to the metal cadmium (Cd) is widespread globally because it is ubiquitously present as a contaminant in nearly all food types, notably the staple food groups [1]. Rice is a staple for half of the world’s population, and it forms half or more of total oral Cd exposure in many regions. A recent finding that 15% of arable soils around the world are loaded with toxic metals, Cd included, raises concern about dietary Cd exposure guidelines and its permissible levels in foods [2]. The European Food Safety Authority (EFSA) set the maximum level (ML) in rice at 0.2 mg/kg [3,4], while Codex’s ML for Cd in rice is as high as 0.4 mg/kg dry grain weight (http://www.fao.org/fileadmin/user_upload/livestockgov/documents/1_CXS_193e.pdf) (accessed 20 June 2025).

Cd has a high toxicity potential, its electronegativity is close to zinc, and its ionic radius is within the same range as calcium; thus, it can enter the body through the metal transporters and pathways for the essential metals, iron [5], zinc [6], copper [7], and calcium [8,9]. No excretory mechanism exists for Cd [10,11]. Acquired Cd is stored by hepatocytes and the proximal tubular cells (PTCs) of the kidneys. CdMT is released when these cells are injured or die [1,12]. Thus, Cd excretion can be viewed as a manifestation of current tissue injury, and reducing present and future exposure cannot mitigate injury in progress [1,12]. The half-life of Cd in the body ranges from 7.4 to 30 years; the lower the bodily burden, the longer the half-life of Cd [13,14,15].

Cells with high metabolic rates, like PTCs, are particularly susceptible to Cd effects on mitochondria [16,17,18]. There, it reduces the synthesis of adenosine triphosphate (ATP), disrupts the electron transport chain, and promotes the formation of reactive oxygen species (ROS), leading to mitochondrial injury, and the release of mitochondrial DNA. In response, the nuclear factor-kappaB (NF-κB) and the DNA-sensing mechanism, cyclic GMP-AMP synthase–stimulator of interferon genes (cGAS-STING), signaling pathways are activated and proinflammatory cytokines are released, followed by premature cell death, inflammation, and tubulointerstitial fibrosis [1]. Also, Cd cytotoxicity targets the nucleus, lysosome, and endoplasmic reticulum [19,20,21]. Many other molecular mechanisms have been postulated to explain the nephrotoxicity of Cd [22,23,24].

The present review provides comprehensive knowledge on Cd exposure and its implications for kidney function and hypertension, with a specific focus on the mechanistic role of the eicosanoid 20-hydroxyeicosatetraenoic acid (20-HETE) and gender-differentiated effects. It integrates epidemiological evidence with molecular insights and critically evaluates current toxicological benchmarks. The central working hypothesis for Cd-induced hypertension via it effects on the kidneys is depicted in Figure 1.

## 2. Food as a Modifiable Cd Exposure Route

This section highlights food safety monitoring programs, known as total diet studies and market basket surveys. In addition, the vulnerable groups of people who have enhanced Cd uptake rates are briefly discussed.

### 2.1. Exposure Limits

Reliable data on the concentration of Cd in various food items and estimated dietary exposure levels can be found in reports of total diet studies [25,26,27,28]. Because Cd is notoriously known for its toxicity to kidneys and bones, permissible exposure levels, namely the tolerable intake level, the minimal risk level, the toxicological reference value, and the reference dose, have been determined for these targets [29]. Like the ML values of Cd in rice, there is no consensus on exposure limits, nor are there thresholds among the existing exposure guidelines. Notably, however, most countries adopt a Cd exposure of 0.83 μg/kg bw/day (58 μg/day for a 70 kg person) and Cd excretion rate of 5.24 µg/g creatinine as exposure threshold levels. These Cd exposure guidelines were determined by the Joint Food and Agriculture Organization and World Health Organization Expert Committee on Food Additives and Contaminants (JECFA) (https://apps.who.int/iris/handle/10665/44521) (accessed on 20 June 2025).

### 2.2. Populations at Risk of Dietary Cd Exposure and Adverse Health Effects

Young children are at risk because of their enhanced intestinal absorption of metals and larger food intake relative to body size, and thus their larger dose of food-borne toxicants [30,31]. Excessive Cd exposure early in life was studied in the U.S.; the mean and 90th-percentile Cd exposures for children aged 1–6 years, estimated from total diet study data from 2018 to 2020 and food consumption data from 2015 to 2018, were 0.43 and 0.71 μg/kg bw/day, respectively. The food groups contributing most to Cd exposure were grains/baking, dairy and fruit and grains/baking and vegetables, respectively [28]. These dietary Cd exposure levels exceeded the toxicological reference value (TRV) for oral Cd exposure (0.21–0.36 μg/kg bw/day) [32], but they were within the JECFA’s tolerable daily intake level of 0.83 μg/kg bw/day. The discrepancies in existing Cd exposure limits need to be addressed.

Excessive Cd exposures early in life may adversely affect the kidneys. This has been observed in a prospective cohort study of rural Bangladeshi children, where Cd exposure was inversely associated with eGFR and a reduced kidney volume, particularly in girls [33]. The eGFR values in Mexican children, aged 8–12 years, varied inversely with eGFR [34].

### 2.3. A Broad Range of Adverse Health Effects of Cd

Systematic reviews and meta-analyses have suggested that the risks of diabetes [35], cardiovascular disease (CVD) [36], chronic kidney disease (CKD) [37], and hypertension [38] rise with blood Cd levels and/or urinary Cd excretion rates, in a dose-dependent manner, as do mortality from any cause and CVD [39]. Furthermore, a Dutch prospective cohort study showed that Cd promoted the progressive deterioration of eGFR in patients with diabetes [40]. A fall in eGFR at a high rate in Swiss cohort participants was linked with Cd exposure [41]. Notably, in the mentioned meta-analyses, effects of Cd were found to be independent of traditional risk factors, like adiposity. In a review of Cd and diabetes, Cd exposure was inversely associated with BMI and other measures of obesity in children, adolescents, and adults [42].

It can be inferred from the above findings that a significant proportion of people have been affected by their exposure to Cd in a normal diet. This calls into question whether safe exposure levels exist for a cumulative toxicant like Cd. The existing permissible dietary exposure levels and exposure thresholds are outdated, as is the utility of β_2_-microglobulinuria (β_2_M-uria), defined as excretion of β_2_M at a rate of 300 µg/g creatinine, to reflect the nephrotoxicity of Cd [29]. A further discussion on the use of eGFR decline instead of β_2_M-uria in determining the critical exposure level for Cd is provided in Section 4.

## 3. Cd Exposure and Hypertension

This section defines and describes the prevalence of hypertension, together with CKD diagnosis and staging and cardiovascular–kidney–metabolic (CKM) syndrome, proposed by the American Heart Association in view of the high prevalence of metabolic and kidney disease. Cd exposure levels that may increase risk of hypertension are summarized. High blood pressure as a mediator of Cd-induced albuminuria is discussed.

### 3.1. Hypertension Prevalence

Hypertension is defined as systolic blood pressure (SBP) and/or diastolic blood pressure (DBP)  ≥ 140/90 mm Hg [43,44]. In most economically developed countries, at least one-third of their adult population live with primary or essential hypertension, with notable gender differences in incidence and treatment outcomes [45,46].

In a U.S. prospective cohort study, the Health and Retirement Study (1992–2014), hypertension occurred in 41% of cohort participants over a median observation period of 7.8 years, and the estimated incident hypertension was 45.3 per 1000 person-years [47]. A cross-sectional study from France (the CONSTANCES study) reported a higher prevalence of hypertension in men (37.3%) than women (23.2%), with body mass index (BMI) being a strong determinant, especially in women [48]. In rural Bangladesh, hypertension was found to be more prevalent in women (8.9%) than men (4.5%), and the highest prevalence (21.3%) was found in women aged ≥ 60 years [49]. Risk factors for hypertension in rural residents of Bangladesh included age, education, current tobacco use, BMI, and hyperglycemia [49].

### 3.2. Resistance Hypertension: An Emerging Challenge

Use of antihypertensive medications, which include diuretics, dihydropyridine calcium channel blockers, and renin–angiotensin system inhibitors fails to achieve target blood pressure control in 10–15% of hypertensive patients [50,51]. The condition is designated as resistant hypertension. Risk factors for resistant hypertension are age, high blood pressure, obesity, salt consumption, CKD, and diabetes mellitus (DM) [50,51]. Notably, hypertension and DM are leading causes of end-stage kidney disease, and kidney damage is a pathological sign common among those with hypertension and DM. Those who had hypertension onset at <35, <45, and ≥45 years of age had, respectively, 2.52-fold, 1.59-fold, and 1.54-fold higher odds of CKD [52].

As listed in Table 1, the diagnosis and staging of CKD are based on eGFR reduction and albuminuria [53].

Lower eGFR and higher albumin excretion rates are both predictors of death from any cause and CVD, independent of each other and of traditional risk factors for CVD [54]. Because of the high prevalence of metabolic and kidney disease, the American Heart Association has put forward the CKM syndrome and sex-specific, race-free risk equations: PREVENT (AHA Predicting Risk of CVD Events) [55,56,57].

### 3.3. Cd as a Risk Factor for Hypertension and CKD: Epidemiological Data

Table 2 provides Cd exposure levels that have been found to be associated with the risk of hypertension and other adverse effects in environmentally exposed populations.

In repeated measurements of exposure to metals and kidney injury biomarkers [62], Yin et al. (2024) applied pathway enrichment analysis to identify the molecular entities that may account for an association between an increased risk of CKD and exposure to toxic metals; Cd, Cr, and Pb. They connected metal-induced CKD onset to the oxidative stress pathway and a decrease in the expression of the transcription factor NFE2-like BZIP transcription factor 2 (NFE2L2), also known as nuclear receptor factor 2 (NRF2).

### 3.4. Albuminuria in Cd-Exposed People

#### 3.4.1. Tubular Handling of Albumin

Albumin is synthesized in the liver and secreted into the circulation at 10–15 g per day [65,66]. It undergoes catabolism in muscle, the liver, and kidneys, which balances synthesis, and homeostasis is maintained. The normal plasma albumin concentration is between 3.5 g/dL and 5 g/dL, and the half-life in plasma is 19 days [65,66]. With a molecular weight of 66 kDa and negative charge, albumin is not filtered by glomeruli. By means of transcytosis through endothelial cells and podocyte foot processes, 1–10 g of albumin enters the urinary space each day [65,66,67,68].

As Figure 2 depicts, reuptake of albumin and β_2_M is through megalin/cubilin-mediated endocytosis, expressed on the apical membrane of PTCs. Most albumin that reaches tubular lumen is reabsorbed via the high-capacity fluid phase endocytosis and is returned to blood supply via transcytosis [67,68,69]. This albumin reuptake mechanism occurs in S1, S2 and S3 segments of the PT. A relatively small proportion of albumin is reabsorbed via the high affinity, low-capacity (megalin/cubilin) mediated endocytosis, followed by lysosomal degradation [67,68,69]. This mechanism occurs in S1 segment of tubules.

Given that β_2_M-uria is used as a criterion to judge the critical effect of Cd on kidneys, the most frequently reported sign of the nephrotoxicity of Cd is β_2_M-uria, detailed in Section 4. In comparison, the pathogenesis of albuminuria in Cd-exposed people has rarely been investigated. Herein, studies examining albumin excretion in two Cd exposure scenarios are recapitulated.

#### 3.4.2. Moderate-to-High Exposure to Cd

In a study of 519 Thai subjects with moderate-to-high exposure (geometric mean for Cd excretion at a rate of 5.44 µg/g creatinine), estimated fractional reductions in albumin and β_2_M reabsorption were 18 and 21%, assuming glomerular sieving coefficients for albumin and β_2_M of 10^−4^ and 0.01, respectively [70]. Cd may adversely affect megalin, a component involved in the uptake of both albumin and β_2_M [70,71]. In theory, a change in fractional clearance of any filtered protein is a result of filtration and reuptake [69].

As Figure 2 depicts, “toxic” unbound Cd can be released as albumin is degraded [72,73]. Free (unbound) Cd can reach the inner membrane of mitochondria, where it causes excessive ROS production, membrane lipid peroxidation, and mitochondrial DNA leakage [16,19]. As a result, the DNA-sensing mechanism (cGAS-STING) and NF-κB signaling pathways are stimulated and proinflammatory cytokines are released, followed by cell death.

Initially, the stimulator of interferon gene (STING) was found to mediate innate immunity via a DNA-sensing pathway. Its involvement in ferroptosis-induced tubular cell death was demonstrated using an experimental model of ischemic kidney injury [74]. Intriguingly, the potential contribution of the immune system to modulating blood pressure variability and renal injury has emerged [75] as is the role of proximal tubule endocytosis in renal fibrosis development [76].

#### 3.4.3. Low-to-Moderate Exposure to Cd

Among 641 study subjects with low-to-moderate exposure to Cd, evident from a geometric mean for Cd excretion of 1.11 µg/g creatinine, hypertension was the most prevalent (39.8%), followed by albuminuria (16.5%) and a low eGFR (4.8%) [77]. It is inferred from a simple mediation model analysis that the albumin excretion rate in this group is causally related to an increase in SBP, which accompanies Cd-induced eGFR reductions. The direct relationship between the albumin excretion rate and blood pressure is shown in Figure 3.

The potential effect of blood pressure on the albumin excretion rate was evident from the scatterplots, where the albumin excretion rate varied directly with SBP and DBP (Figure 3a,b). Although the R^2^ values are relatively low (e.g., 0.059–0.088), the associations are statistically significant (*p* < 0.01). These low R^2^ values are not uncommon in population-based studies, where multiple confounding factors may influence outcomes. The higher R^2^ value in the large sample size (*n* = 238) than the modest sample size (*n* = 162) supports the observed trends.

Those with SBP ≥ 130 mm Hg plus low eGFR had 17.2% and 21.2% higher albumin excretion rates compared to those with eGFR 61–90 and ≥90 mL/min/1.73 m^2^, respectively (Figure 3c). Similarly, those with high DBP and low eGFR had 19.0% and 22.5% higher albumin excretion rates compared to those with eGFR 61–90 and ≥90 mL/min/1.73 m^2^, respectively (Figure 3d). Thus, there was an increase in albumin excretion in those with low eGFR without hypertension (SBP/DBP of 130/80 mm Hg).

The above observations were in line with data from the SPRINT Trial, where participants who had low eGFR and a 16 mm Hg higher SBP than the mean SBP were found to have an albumin excretion rate rise by 16% [78]. Proteinuria/albuminuria are predictors of a continued progressive decline of eGFR [79,80,81,82].

### 3.5. The Kidney and Gender Differences in Cd-Induced Hypertension

#### 3.5.1. Cd and Kidneys’ Role in Blood Pressure Regulation

The essential role of the kidneys in long-term blood pressure control was first demonstrated in 1970s, when the transplant of a kidney from a hypertensive rat to a normotensive host raised blood pressure, and the transplant of a normal kidney to a hypertensive host lowered blood pressure [83,84,85]. The close relationships between renal perfusion pressure, urine flow, and sodium excretion led to Guyton’s theory that hypertension develops when the pressure-natriuresis response is shifted to higher pressure levels [86]. Thus, the regulation of sodium transport in kidneys by blood pressure is central to Guyton’s model of blood pressure control [87]. Many mediators of the pressure-natriuresis response have been investigated, such as 20-HETE [88,89], renal sympathetic activity [90], and the renin–angiotensin–aldosterone system (RAAS) [91].

Herein, the role of 20-HETE in regulating the salt balance in the kidneys is discussed, together with its potential involvement in Cd-induced hypertension. This eicosanoid is produced from arachidonic acid via ɷ-hydroxylation reaction, catalyzed by CYP4A11 and CYP4F2 enzymes [92]. Evidence that links these enzymes to hypertension comes from human gene mutation research, along with 20-HETE deficiency, which causes the retention of sodium and development of hypertension in Dahl salt-sensitive rats [88].

Below is a hypothetical model for a Cd-intoxicated kidney tubular cell (Figure 4).

#### 3.5.2. The Increment of Tubular Avidity for Filtered Sodium After Cd Exposure

In the 1990s, the synthesis and excretion of 20-HETE by human kidneys were observed [95,96]. Later, CYP4F2 and CYP4A11 were found to be responsible for its synthesis [90]. CYP4F2 was found to be more prominently expressed in glomeruli and proximal tubules than the distal tubules [97]. In comparison, CYP4A11 was faintly expressed in the proximal and distal tubules, but not in glomeruli [97]. Notably, the zinc influx transporter, ZIP8, known to mediate Cd uptake, was more prominently expressed in the distal than the proximal tubules, but not in the glomeruli [98]. The Na/K-ATPase is expressed in the distal tubule in the highest density, especially in the basolateral membrane [94,99].

The activity of Na/K-ATPase pump was highly sensitive to oxidative damage [100], and it is subjected to degradation by the proteasome and endo-lysosomal protease following oxidative damage [100]. This decreases the cellular Na/K-ATPase content and sodium transport activity. In the cortical proximal tubular cell, where there are relatively higher Cd accumulation levels than the distal tubules, an increased CYP4F2 expression was noted [101,102,103]. However, because of a low abundance of Na/K-ATPase due to Cd-induced oxidative damage, the ability of 20-HETE to inhibit salt reabsorption is impaired.

Low CYP4A11 protein expression in the distal tubules in those with an elevated Cd bodily burden, as suggested by an inverse correlation between kidney Cd levels and CYP4A11 protein abundance [103], suggests that only low-level 20-HETE is produced in this nephron region, where 20-HETE reduces sodium reabsorption through inhibition of Na+-K+-2Cl− and 70pS K+ transport. Low 20-HETE levels in the distal tubule thus lead to a greater reuptake of salt and water into systemic circulation. Hypertension develops after prolonged volume overload, increasing sodium retention and shifting the natriuretic response to higher pressures. Rats with Cd-induced hypertension showed increased sodium retention and reduced sodium excretion [104,105].

#### 3.5.3. Gender Differences in Cd-Induced Hypertension

The female preponderance of Cd-associated hypertension has been attributed to an effect of Cd on sex hormones. Japanese and Swedish studies on women showed that serum estradiol levels were inversely associated with the Cd excretion rate [106,107]. Another study on postmenopausal Japanese women observed that serum testosterone levels rose with the Cd excretion rate [108]. In the U.S. NHANES 2013–2016 dataset, hyperandrogenemia, assessed with the free androgen index and a ratio of total testosterone to sex hormone-binding globulin ≥5%, was associated with exposure to Cd, especially in women aged ≥ 35 years [109]. When comparing blood Cd in the top tertile to the bottom, it was found that the risk of hyperandrogenemia rose by 20%. Inflammation associated with Cd was suggested to mediate the effect observed.

Clinical and experimental data indicate that an effect of androgen on blood pressure is mediated by 20-HETE [109,110,111,112]. In a study on Thai women, hypertension and Cd excretion were associated, respectively, with 1.9-fold and 4.4-fold increases in the odds of urinary 20-HETE concentrations higher than the median value of 469 pg/mL [113]. Also, SBP correlates positively with urine 20-HETE, and a rise of 20-HETE excretion from the bottom to the top tertile was associated with 6 mm Hg higher SBP after adjustment of covariates. These data suggest that Cd may affect 20-HETE synthesis via the tubular cells, consistent with immunoblotting data [101,102,103].

## 4. Critical Exposure Levels of Cd

This section highlights the critical Cd excretion rate, obtained by applying the advanced benchmark dose (BMD) methodology. Special emphasis is given to the BMD modeling of dose–effect datasets from population samples, which could be representative of the general populations with low to moderate environmental Cd exposure. It recapitulates the results from a recent investigation, suggesting the use of eGFR decline, as opposed to β_2_M excretion, in Cd toxicological risk assessment. Also, it underscores the total imprecision in measuring Cd exposure and its associated adverse effects, which obscures the relationship between eGFR and Cd excretion. In some instances, an effect of Cd is nullified, with the phenomena described in the literature as “reverse causality”.

### 4.1. Benchmark Dose Limit (BMDL)

The benchmark dose (BMD) of any exposure indicator is obtained by fitting the entire exposure–effect dataset to a mathematical dose–response model, in which a specific effect size, known as the benchmark response (BMR), is pre-defined [114]. The difference between the lower bound (BMDL) and upper bound (BMDU) of the 95% confidence interval (CI) of the BMD reflects the statistical uncertainties in the BMD estimates. A narrow difference indicates a high degree of certainty of the estimated BMD figures. A wide difference, e.g., a BMDU/BMDL ratio ≥ 100, is indicative of unreliable BMD estimates.

The mathematical dose–response models often used are inverse exponential, natural logarithmic, exponential, and Hill models [115,116]. To evaluate how well each dose–response model fits the data, the Akaike information criterion (AIC) is used [115,116]. The AIC assesses for both goodness of fit and the complexity of the model. The model weight is measured relative to an amount of information lost by a given model and it provides an additional insight into the shape and steepness of the slope [117,118].

The BMDL value of any exposure indicator is defined as the lower 95% confidence bound of the BMD, computed at a 5% BMR. This BMDL value has replaced the no-observed-adverse-effect level (NOAEL) and can represent a critical exposure level [114].

### 4.2. JECFA and EFSA Dietary Cd Exposure Guidelines and Thresholds

The JECFA assumes that a lifetime exposure to 2 g of Cd is tolerable and that β_2_M excretion at a rate of 300 µg/g creatinine represents a critical effect of Cd. It derived the tolerable Cd dose of 0.83 µg/kg bw/day and assigned a Cd excretion rate at 5.24 µg/g creatinine as a threshold. With an application of the same β_2_M endpoint, the European Food Safety Authority (EFSA) found a dietary Cd exposure at 0.36 μg/kg bw/day to be the reference dose (RfD), and a Cd excretion rate at 1 µg/g creatinine was assigned as the threshold level after a safety margin was included to account for interindividual variation in the exposure levels of Cd [3,4].

A conventional dosing experiment reported that the RfD value for human Cd exposure was 0.2 μg/kg bw/day [119]. This empirical Cd exposure limit is 24.1 and 55.6% of the Cd exposure limits derived by the JECFA and EFSA, respectively. In a conventional study, pigs were fed with four dose levels of Cd—0, 0.5, 2, 8, and 32 mg Cd/kg feed—for 100 days [119], and the Cd dose producing abnormal β_2_M excretion was the highest, while abnormal excretion of retinol binding protein (RBP) occurred at the lowest feeding dose. These findings that indicate β_2_M excretion was not an early warning sign of Cd nephrotoxicity and its use as a basis to estimate exposure guidelines represents a conceptual flaw.

### 4.3. Falling eGFR as an Early Warning Sign of Cd Nephrotoxicity

As the data in Table 1 indicate, eGFR is a parameter for CKD diagnosis and severity staging. Surprisingly, however, eGFR has never been used in toxicological risk assessment of Cd. The ignorance of the eGFR endpoint could be attributed to the results reported in two meta-analyses that found Cd had no effect on eGFR, nor did it promote progressive reductions in eGFR toward kidney failure among Cd-exposed people [120,121].

Table 3 provides the results of logistic regression analyses evaluating the effects of doubling the Cd excretion rate on the prevalence odds of CKD along with other independent variables.

All independent variables incorporated in models A and B were identical, with the exception of Cd excretion rates. In model A, Cd excretion (E_Cd_) was normalized to creatinine excretion (E_cr_), denoted as E_Cd_/E_cr_. This practice serves to correct for different urine volumes or dilutions among study subjects in the case of single-time (spot) urine sampling. In model B, E_Cd_ was normalized to creatinine clearance (C_cr_), denoted as E_Cd_/C_cr_. This C_cr_ normalization corrects for the variability in both urine volume and number of surviving nephrons.

In model A, the POR for CKD increased with old age, BMI ≥ 24 kg/m^2^, and E_Cd_/E_cr_. The CKD risk rose 1.98-fold per doubling E_Cd_/E_cr_. In model B, the POR for CKD increased with old age, BMI ≥ 24 kg/m^2^, hypertension, and E_Cd_/C_cr_. Odds of CKD rose 3.13-fold per doubling E_Cd_/C_cr_, and 2.6-fold in those with hypertension.

In summary, the risk of CKD was reduced by 63% when comparing E_cr_ vs. C_cr_ normalization, while the effect of hypertension on the CKD risk was completely obscure. These data clearly demonstrate an impact of E_cr_ adjustment because it is influenced by muscle mass, which varies among people and genders (universally higher in males than females). Adjusting E_Cd_ with E_cr_ adds variance to the datasets, which can diminish or nullify an effect size [123]. C_cr_ normalization is unaffected by muscle mass, and it can be computed by an equation, thereby obviating timed urine collection [124]. It is recommended that E_Cd_ should be normalized to C_cr_ where possible, instead of E_cr_.

### 4.4. Misuse of β_2_M Excretion to Indicate the Tubular Effect of Cd

Most nucleated cells in the body express on their surface the protein β_2_M, a component of class I histocompatibility complexes, which can be released into the blood circulation [125,126]. With the molecular weight of 11,800 Daltons, β_2_M is filtered totally with plasma by the glomeruli, reabsorbed by PTCs in the S1 segment of tubules through endocytosis, mediated by the megalin/cubilin receptor system, and degraded in the lysosome (Figure 5). Unlike albumin, which is taken up and returned to the circulation, there is little evidence that retrieved β_2_M is re-entered into the blood supply.

Excretion of proteins with low molecular weights, like β_2_M, α_1_-microglobulin (α_1_M), and retinol binding protein (RBP), has long been used as a sign of Cd-induced tubular dysfunction [128]. Such utility relies on the promise that they are completely filtered by glomeruli and reuptaken by kidney tubules. In theory, excretion of these proteins is a function of their production, glomerular filtration, reuptake, and degradation by PTCs.

A recent analysis of β_2_M homeostasis determinants has unveiled for the first time that fractional tubular degradation of filtered β_2_M (FrTD_β2M_) should be used in the assessment of the tubular effect of Cd [127]. A large interindividual variation in the flux of β_2_M from cells into plasma means that β_2_M excretion can only be minimally related to tubular degradation of β_2_M. Consequently, the excretion of β_2_M cannot be a reliable measure of tubular function [128].

The fact that excretion of β_2_M, especially among those with low-dose exposures, does not reflect the tubular effect of Cd is evident, as it was when BMD modeling was applied to data from 799 participants of a Thai cohort of non-diabetics, with a mean age of 49.2 years (range: 18–87) [122]. Based in the geometric mean of Cd excretion of 2.15 µg/g creatinine (1.82 µg/L filtrate), this group can be considered to represent low-to-moderate exposure observed in most populations. The BMDL values of Cd excretion were based on changes in eGFR and the β_2_M excretion rate. The BMR was set at 5% for both endpoints.

Figure 6 provides the results of correlation analysis, while Figure 7 shows the outputs from the PROAST software for BMD computation (https://proastweb.rivm.nl) (accessed on 25 June 2025).

As Figure 6 depicts, the strengths of the correlations of eGFR vs. Cd excretion rate and β_2_M excretion rate vs. eGFR were higher when E_Cd_ and E_β2M_ were normalized to C_cr_, compared to E_cr_-adjusted datasets.

As Figure 7 depicts, Cd excretion of 0.17 µg/g creatinine was found to be a critical Cd exposure level (BMDL), when a decrease in eGFR was a toxic endpoint. The BMDU/BMDL ratio for this endpoint was 16.9, meaning a high degree of certainty of the BMD estimates. However, when β_2_M excretion was a toxic endpoint, the BMDL value of the Cd excretion rate could not be determined because of a high degree of uncertainty around the BMD estimates (the BMDU/BMDL ratio = 182). These findings suggest that an increase in excretion of β_2_M in people exposed to low-dose Cd is not statistically related to Cd excretion. It appears, therefore, that there is little basis for using β_2_M excretion as the indicator of the Cd effect and to derive exposure guidelines.

β_2_M-nuria, defined as a β_2_M excretion rate of 300 µg/g creatinine, and its Cd excretion threshold of 5.24 µg/g creatinine are the manifestations of nephron destruction due to the nephrotoxicity of Cd and high blood pressure (hypertension) (Table 3). A Cd excretion rate as little as 0.17 µg/g creatinine is an early warning sign of Cd nephrotoxicity suitable for derivation of the Cd exposure level below which kidney functional integrity is preserved.

A small increase in excretion of β_2_M has been found to be associated with an increased risk of hypertension independently of albuminuria in the general Japanese population [129,130]. Two studies conducted on Cd-exposed subjects suggested β_2_M excretion to be a risk factor for hypertension, especially among those with diabetes [131,132]. These findings are consistent with the notion that β_2_M may be involved in blood pressure control [133,134,135]. An elevated plasma β_2_M was found to be associated with increases in prevalent and incident hypertension in the Framingham Heart Study [135].

## 5. Conclusions

The prevalence of hypertension, diabetes type 2, CVD, and CKD all increased with blood Cd concentrations and/or Cd excretion rates [35,36,37,38,39], indicative of the presence of Cd and its accumulation in the human body. These disease prevalence increments are independent of the traditional risk factors, obesity and tobacco consumption. These data together with the current scientific understanding of the nephrotoxicity of Cd at the cellular and molecular levels are indisputable.

Exposure to Cd at the levels presently found in a normal diet has posed a significant public health threat. Toxic levels of Cd can be found in the general population at a proportion exceeding the 5% public health standard for environmental diseases. Notably, however, such an impact of dietary Cd exposure on population health has been largely underappreciated and ignored. Now is the time to recognize the health threat of Cd and the need to address key issues of toxic risk assessment criteria.

Hypertension is a widely recognized sequala of kidney damage, which is associated frequently with the renin–angiotensin system of blood pressure control. Herein, it was shown that 20-HETE may contribute to an enhanced risk of hypertension following Cd exposure through an increased tubular avidity for filtered sodium. This finding explains the higher risk of Cd-induced hypertension in women compared to men; Cd-related elevations in plasma testosterone (hyperandrogenemia) and urine 20 HETE excretion were found only in women [107,108,109,113]. Evidence that 20-HETE contributes to blood pressure homeostasis comes from human gene mutation research and in vitro and animal studies. Investigation into the 20-HETE roles in the gender differential effects of Cd is limited. Further molecular mechanistic dissections, and the localization CYP4F2 and CYP4A11, which produce 20-HETE, along the nephron are warranted. These studies will pave the way for novel dietary interventions and therapies, especially for resistance hypertension.

CKD in its early stage is largely asymptomatic. This makes its early detection difficult and the initiation of early treatment, which can significantly prevent progression to kidney failure, limited. In low-to-moderate Cd exposure scenarios, albuminuria is at least three times more prevalent than a low eGFR. Thus, it is suggested that an increase in albumin excretion can be used to find those with early CKD. Proteinuria can also be used for such purposes because the BMDL value of Cd excretion for the proteinuria endpoint is as little as 0.05 µg/g creatinine [136]. This could form an early warning sign of Cd nephrotoxicity.

Albumin excretion within the normal range was independently associated with a higher risk of hypertension in the general population [137]. When comparing with the lowest category of albumin excretion, the risk was increased 1.75-fold in the highest albumin excretion group. The hypertension risk associated with an increase in albumin excretion was particularly high in women; the risk of hypertension was 2.47-fold higher compared to those with the group with the lowest albumin excretion [137]. Albumin excretion at a rate of 7 mg/g creatinine predicted incident CKD within 10 years [138]. Albumin excretion at a rate of 10 mg/g creatinine may lead to death from all causes and CVD [139].

When using the eGFR as an endpoint, a Cd excretion rate of 0.17 µg/g creatinine was found to be the lowest Cd excretion rate associated with a 5% reduction in eGFR. In contrast, the Cd excretion rate associated with a 5% increase in β_2_M excretion could not be reliably determined. A large interindividual variability in plasma β_2_M concentrations means that the β_2_M excretion cannot be precisely related to tubular function. Consequently, the BMDL value of Cd excretion could not be identified when β_2_M excretion was an endpoint (Figure 7). Because a reduction in eGFR is one of the diagnostic criteria for CKD, Cd excretion at a rate of 0.17 µg/g creatinine could serve as the basis for computing an exposure limit for Cd. This figure is 3.2% of the threshold level that was derived by the JECFA with β_2_M-nuria as an endpoint.

Environmental Cd exposure in many parts of the world has now reached toxic levels in a significant proportion of people. No consensus on Cd exposure limits exists nor its permissible levels in food and water. The Australia and New Zealand Standard for Cd in drinking water is 2 µg/L, lower than the 5 µg/L of the WHO drinking water guidelines. Current evidence also suggests that the eGFR deterioration due to Cd-induced nephron destruction is irreversible. Health-protective exposure guidelines should be established for Cd. Therapeutically effective chelation treatment to remove Cd from the kidneys does not exist. Essential preventive measures should include avoiding smoking and foods containing high Cd levels, keeping an optimal body weight, and minimizing Cd assimilation and the kidney burden of Cd by maintaining optimal body contents of calcium, zinc, and iron.

## Figures and Tables

**Figure 1 jox-15-00122-f001:**
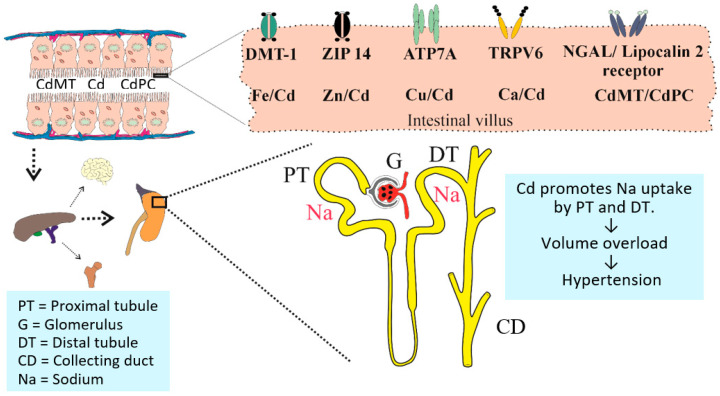
Cadmium absorption and transport to the nephron. Cd from foods reaches systemic circulation via the portal blood system and the liver. The S1 segment of proximal tubule (PT) takes up Cd from the glomerular filtrate, stores it in complexes with metallothionein (CdMT), which are released following injury or death of the PT cells. Cd may also reach the distal tubule (DT). The PT and DT are the nephron regions responsible for salt and water balance that maintains blood pressure homeostasis. Prolonged Cd exposure may increase Na uptake by PT and DT, leading to volume overload and hypertension. The specialized transport proteins involved in the intestinal absorption of Cd are DMT-1, ZIP14, ATP7A, TRPV6, and NGAL/lipocalin 2 receptor.

**Figure 2 jox-15-00122-f002:**
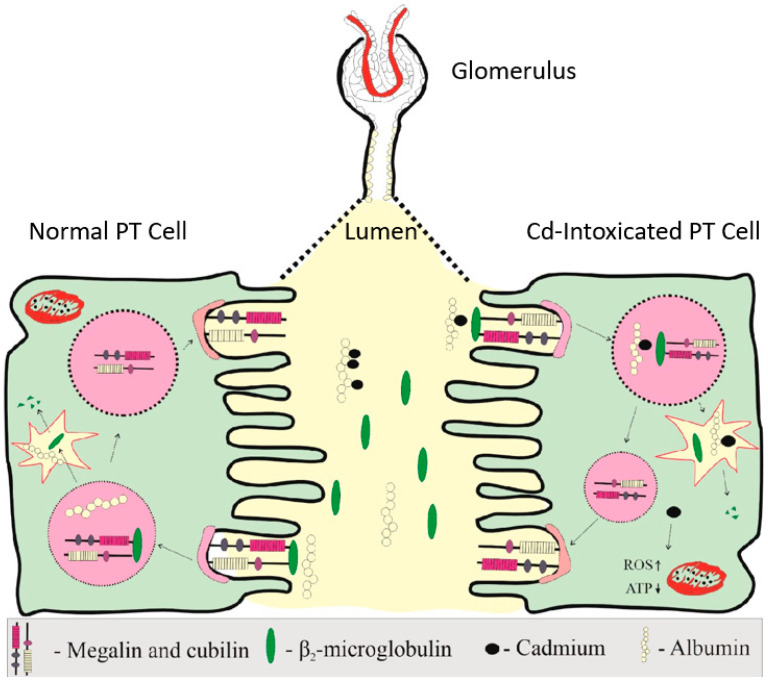
Albumin and β_2_-microglobulin reuptake by the proximal tubular cells (PTCs). Albumin reaches tubular lumen through endothelial cells and podocyte foot processes [65,66], and nearly all of it is reabsorbed and returned to blood circulation by fluid-phase endocytosis and transcytosis [67,68]. A relatively small proportion of albumin is reabsorbed by megalin/cubilin-mediated endocytosis and undergoes lysosomal degradation. In Cd-intoxicated PTC, “toxic” unbound Cd is released as albumin is degraded.

**Figure 3 jox-15-00122-f003:**
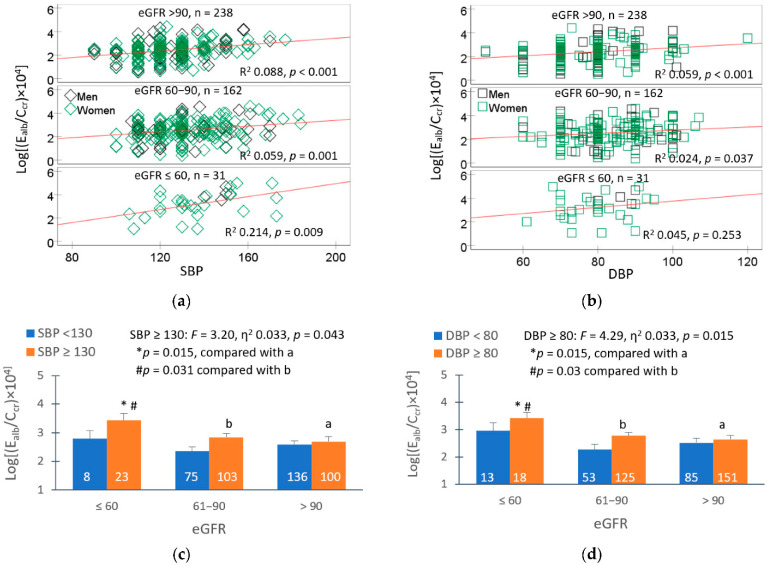
Effects of blood pressure and eGFR on albumin excretion. Scatterplots relate albumin excretion to SBP (**a**) and DBP (**b**) across eGFR groups; ≤60, 61–90, and ≥90 mL/min/1.73 m^2^. Mean albumin excretion rate in eGFR subgroups with SBP (**c**) and DBP above medians (**d**). The median SBP/DBP was 130/80 80 mmHg. All mean values were adjusted for covariates and interactions. The diamonds and squares represent male and female participants. Data are from Satarug et al. https://doi.org/10.3390/toxics13020081 (accessed on 20 June 2025) [77].

**Figure 4 jox-15-00122-f004:**
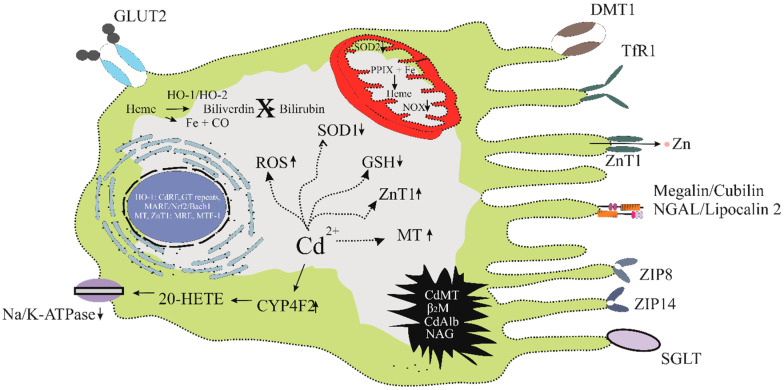
The kidney tubular cell as Cd target of toxicity. Cd enters tubular cells and reaches mitochondria through receptors and transporters for essential metals. It also targets the lysosome, endoplasmic reticulum, and nucleus. It induces expression of CYP4F2, which produces 20-HETE. It enhances mitochondrial production of ROS, which can damage Na/K-ATPase and apical Na-cotransporters. The electrochemical Na gradient generated by the Na/K-ATPase pump promotes co-transport of Na with other filtered substances, notably glucose [93,94]. The molecular targets of Cd are Na/K-ATPase, CYP4F2, MT, ZnT1, GSH, ROS, SOD1, SOD2, NOX, de novo heme, and bilirubin syntheses. In the nucleus, Cd targets the expression of numerous genes through the GT repeats in the promoter region of heme oxygenase-1, the Cd response element, the metal response element (MRE), the metal response element-binding transcription factor-1 (MTF-1), and MARE/NfF2/Bach1.

**Figure 5 jox-15-00122-f005:**
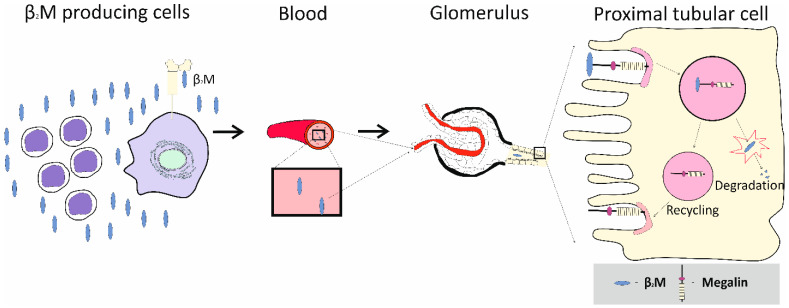
Fate of β_2_-microglobulin in the body. The protein β_2_M is shed into bloodstream by nucleated cells. By virtue of its small mass, β_2_M is filtered completely by the glomeruli, retrieved by PTCs, and subjected to lysosomal degradation. Figure is from Phelps et al. https://doi.org/10.20517/scierxiv.2025.60.v1 (accessed on 20 June 2025) [127].

**Figure 6 jox-15-00122-f006:**
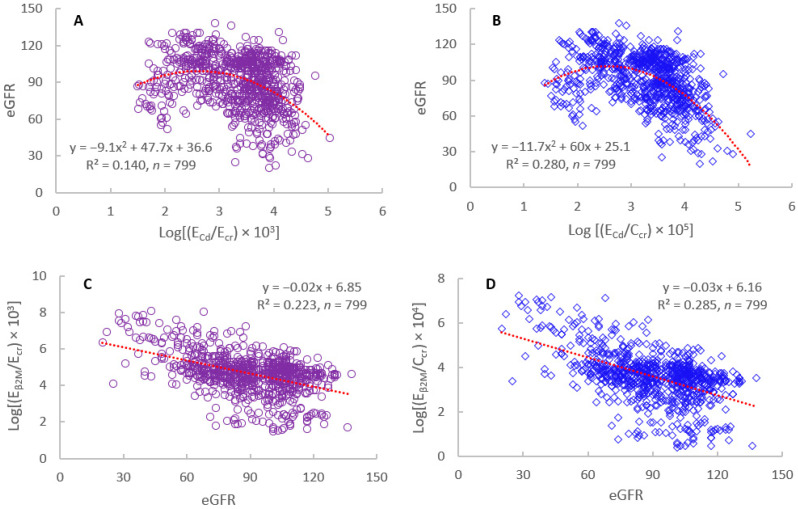
Inverse relationship of eGFR vs. E_Cd_ and E_β2M_ vs. E_Cd_. Scatterplots relate eGFR to E_Cd_/E_cr_ (**A**) and E_Cd_/C_cr_ (**B**). Scatterplots relate E_β2M_/E_cr_ (**C**) and E_β2M_/C_cr_ (**D**) to eGFR. Data are from Đorđević et al. https://doi.org/10.21203/rs.3.rs-6799604/v1 (accessed on 20 June 2025) [122].

**Figure 7 jox-15-00122-f007:**
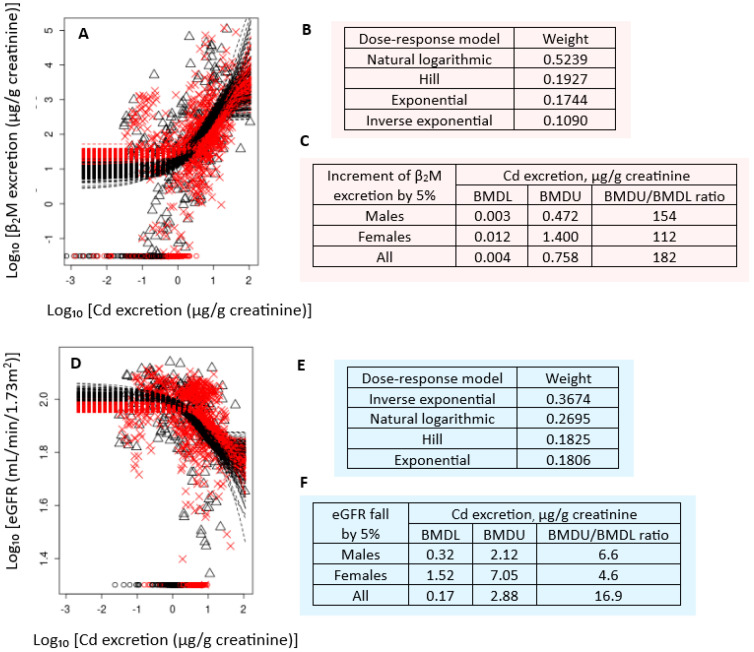
BMDL values of Cd excretion computed from eGFR and β^2^M endpoints. Bootstrap dose–effect averaging for β_2_M excretion−Cd excretion (**A**) and eGFR−Cd excretion (**D**). The mathematical dose–response models and weights for β_2_M excretion−Cd excretion (**B**) and eGFR−Cd excretion (**E**). BMDL and BMDU values of E_Cd_/E_cr_ for the β_2_M excretion (**C**) and falling eGFR (**F**). × and △ represent male and female participants. Data are from Đorđević et al. https://doi.org/10.21203/rs.3.rs-6799604/v1 (accessed on 20 June 2025) [122].

**Table 1 jox-15-00122-t001:** Staging of CKD according to eGFR and albuminuria.

GFR Domain	Albuminuria Domain
G1: Normal or high eGFR ≥ 90 mL/min/1.73 m^2^	A1: Normal to mildly increased AER < 30 mg/d or ACR < 30 mg albumin/g creatinine
G2: Mildly decreased eGFR 60–89 mL/min/1.73 m^2^	A2: Moderately increased AER 30–300 mg/d or ACR 30–300 mg/g creatinine
G3a: Mildly to moderately decreased eGFR 45–59 mL/min/1.73 m^2^ G3b: Moderately to severely decreased eGFR 30–44 mL/min/1.73 m^2^	A3: Severely increased AER > 300 mg/d or ACR > 300 mg/g creatinine
G4: Severely decreased eGFR 15–29 mL/min/1.73 m^2^	
G5: Kidney failure eGFR < 15 mL/min/1.73 m^2^	

eGFR, estimated glomerular filtration rate; AER, albumin excretion rate; ACR, albumin-to-creatinine ratio.

**Table 2 jox-15-00122-t002:** Blood and urine Cd levels associated with adverse health effects in different populations.

Study Population	Findings	Reference
NHANES, 1999–2018 *n* = 38,281, 3.7% resistance hypertension, 27.6% hypertension	Risk of resistant hypertension rose by 30 and 35% when comparing blood Cd in quartiles 3 and 4 with blood Cd in the bottom quartile, respectively.	Chen et al., 2023 [58]
NHANES 1999–2004, *n* = 10,197, ≥20 years	Risk of plasma levels of cardiac troponin (cTnT) ≥ 19 ng/L and of N-terminal pro b-type natriuretic peptide (NT-proBNP) ≥ 125 pg/mL rose by 33 and 39% at blood Cd concentrations ≥ 1.0 μg/L.	Liu et al., 2025 [59]
NHANES 1999–2014 CKD cohort, *n* = 1825, follow-up period, 6.8 years	Risk of all-cause mortality rose by 75 and 59% at Cd excretion rates ≥ 0.60 μg/g creatinine and blood Cd concentrations ≥ 0.70 μg/L, respectively.	Zhang et al., 2023 [60]
Northeast China *n* = 384, four-time repeated measurements, 2016–2021	Cd and Cr produced synergistic effects on NAG excretion, albuminuria, and ACR; Cd and Pb produced synergistic effects on NAG excretion and ACR.	Yin et al., 2024 [61]
Jinzhou, Liaoning, China, *n* = 529, three-time repeated measurements of Cd, Cr, and Pb excretion rates and effects on kidneys, 2016–2021	Baseline median values for urine Cd, Cr, and Pb were 2.41, 3.96, and 2.49 μg/L, respectively. Baseline median values for urine NAG, β_2_M, Alb, ACR, and eGFR were 8.86 U/L, 790 µg/L, 24.4 mg/L, 21.2 mg/g creatinine, and 102 mL/min/1.73 m^2^, respectively. Cd, Cr, and Pb together caused more extensive injury to kidneys than did each individual metal.	Yin et al., 2024 [62]
Korean NHANES 2008–2013 *n* = 40,328, GM for blood Pb and blood Cd in males (females) were 2.5 (1.84) µg/dL and 0.88 (1.04) µg/L	Increases in risk of hypertension by 29, 47, and 78% were associated with Pb, Cd, and combined Cd and Pb exposure, respectively.	Kim et al., 2025 [63]
Korean NHANES 2016–2017 *n* = 4222, ≥30 years, 5.1% had CKD, mean blood Cd 1.2 µg/L	A 2.70-fold rise in risk of CKD was associated with blood Cd in those who had hypertension. A 2.40-fold increase in risk of CKD was associated with blood Cd in non-diabetics.	Yeon et al., 2025 [64]

NHANES, National Health and Nutrition Examination Survey; CKD, chronic kidney disease; Cd, cadmium; Cr, chromium; Pb, lead; NAG, N-acetyl-β-D-glucosaminidase; ACR, albumin-to-creatinine ratio; GM, geometric mean.

**Table 3 jox-15-00122-t003:** Effect of Cd on prevalence odds for chronic kidney disease (GFR domain).

	^a^ CKD			
**Model A**	**POR**	**95% CI**		** *p* **
		**Lower**	**Upper**	
Age	1.173	1.128	1.220	<0.001
Log_2_[(E_Cd_/E_cr_) × 10^3^], µg/g creatinine	1.981	1.500	2.615	<0.001
Gender	1.135	0.533	2.415	0.743
Hypertension	1.933	0.965	3.874	0.063
Smoking	1.140	0.529	2.456	0.738
BMI, kg/m^2^				
12–18	Referent			
19–23	1.150	0.441	3.002	0.775
≥24	4.002	1.351	11.86	0.012
**Model B**	**POR**	**Lower**	**Upper**	** *p* **
Age	1.168	1.119	1.219	<0.001
**Log_2_[(E_Cd_/C_cr_) × 10^5^], µg/L filtrate**	3.132	2.249	4.361	<0.001
Gender	0.719	0.315	1.643	0.434
Hypertension	2.656	1.231	5.727	0.013
Smoking	1.103	0.487	2.495	0.815
BMI, kg/m^2^				
12–18	Referent			
19–23	1.134	0.403	3.189	0.812
≥24	4.784	1.468	15.59	0.009

^a^ CKD was defined as eGFR ≤ 60 mL/min/1.73 m^2^. E_Cd_ was normalized to E_cr_ and C_cr_ in models A and B, respectively. Data were from 691 subjects without diabetes and workplace exposure to Cd (420 females, 271 males), aged 18–87 years [122].

## Data Availability

No new data were created or analyzed in this study.

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
