# Peer review of "Hypertension in People Exposed to Environmental Cadmium: Roles for 20-Hydroxyeicosatetraenoic Acid in the Kidney"

_jox, 2025, doi:10.3390/jox15040122_

Round 1
Reviewer 1 Report
Comments and Suggestions for Authors
This is a clear and well-organized review that explores how environmental cadmium exposure may contribute to hypertension, especially through its effects on the kidneys. The authors do a nice job of pulling together evidence from epidemiologic studies, toxicology, and sex-based differences, offering useful insights into how cadmium might cause kidney and blood vessel damage. The topic is timely and relevant for both environmental and public health, and the manuscript is overall well written and well referenced.
Major and Minor Suggestions:
1. The section on 20-HETE provides valuable insights, but most of the cited evidence comes from animal or in vitro studies. The authors are encouraged to clearly acknowledge the lack of direct human evidence and identify this gap as an important direction for future research.
2. While the R² values in Figure 2 are relatively low (e.g., 0.059–0.088), the associations are statistically significant (p < 0.01). These low R² values are not uncommon in population-based studies, where multiple confounding factors may influence outcomes. A brief explanation in the text would help readers interpret the results appropriately. The large sample size (e.g., n = 238) further supports the observed trends.
3. In Figure 2 panels (a) and (b), please clarify that the diamonds and squares represent male and female participants, respectively.
4. In Figure 6 (A, D), please clarify symbols (e.g., × and △) with a legend, and revise axis labels for clarity: “log₁₀[Cd excretion (µg/g creatinine)]” and “log₁₀[eGFR (mL/min/1.73 m²)]”
This manuscript covers an important and timely topic in environmental health. With minor revisions as noted above, it will be suitable for publication.
Author Response
Reviewer 1.
This is a clear and well-organized review that explores how environmental cadmium exposure may contribute to hypertension, especially through its effects on the kidneys. The authors do a nice job of pulling together evidence from epidemiologic studies, toxicology, and sex-based differences, offering useful insights into how cadmium might cause kidney and blood vessel damage. The topic is timely and relevant for both environmental and public health, and the manuscript is overall well written and well referenced.
Major and Minor Suggestions:
Point 1: The section on 20-HETE provides valuable insights, but most of the cited evidence comes from animal or in vitro studies. The authors are encouraged to clearly acknowledge the lack of direct human evidence and identify this gap as an important direction for future research.
RESPONSE: Thank you for your suggestions to improve a review. In revision, a new Figure 1 detailing the central working hypothesis has been created and added to the text (lines 65-73). Sentences quoted below are added to the Conclusion to stimulate future research (lines 500-504).
“Evidence that 20-HETE contributes to blood pressure homeostasis comes from human gene mutation research, in vitro and animal studies. Studies into its role in gender differential effects of Cd in humans are limited. Further molecular mechanistic dissections in humans are warranted to pave the way to novel dietary interventions and therapies, especially for resistance hypertension.”
Point 2: While the R² values in Figure 2 are relatively low (e.g., 0.059–0.088), the associations are statistically significant (p < 0.01). These low R² values are not uncommon in population-based studies, where multiple confounding factors may influence outcomes. A brief explanation in the text would help readers interpret the results appropriately. The large sample size (e.g., n = 238) further supports the observed trends.
RESPONSE: In a revised version, a new Figure 1 has been added. Thus, an old Figure 2 has become Figure 3. An additional explanation, quoted below, has been provided regarding low R2 (lines 245-251).
“The potential effect of Cd on albumin excretion rate was evident from the scatterplots, where albumin excretion rate varied directly with SBP and DBP (Figure 2a, b). The associations are statistically significant (p < 0.01) although the R² values are relatively low (e.g., 0.059–0.088). These low R² values are not uncommon in population-based studies, where multiple confounding factors may influence outcomes. The higher R2 value in the large sample size (n = 238), compared to the modest sample size (n = 162) lends support to the observed trends.”
Point 3: In Figure 2 panels (a) and (b), please clarify that the diamonds and squares represent male and female participants, respectively.
RESPONSE: The suggestions have been undertaken.
Point 4. In Figure 6 (A, D), please clarify symbols (e.g., × and △) with a legend, and revise axis labels for clarity: “log₁₀[Cd excretion (µg/g creatinine)]” and “log₁₀[eGFR (mL/min/1.73 m²)]”
RESPONSE: Symbols × and △ have been explained. The axis labels have been corrected.
Point 5. This manuscript covers an important and timely topic in environmental health. With minor revisions as noted above, it will be suitable for publication.
RESPONSE: Thank you for your approval.

Reviewer 2 Report
Comments and Suggestions for Authors
This manuscript presents a comprehensive and well-structured review of cadmium (Cd) exposure and its implications for kidney function and hypertension, with a specific focus on the mechanistic role of 20-HETE and gender-differentiated effects. The author successfully integrates epidemiological evidence with molecular insights and critically evaluates current toxicological benchmarks.
The manuscript is scientifically rigorous, clearly written, and of high relevance to environmental health, toxicology, nephrology, and public health policy. Figures and tables are informative, and the conclusions are well-supported by the literature.
Recommendation:
I recommend acceptance.
Author Response
Reviewer 2.
This manuscript presents a comprehensive and well-structured review of cadmium (Cd) exposure and its implications for kidney function and hypertension, with a specific focus on the mechanistic role of 20-HETE and gender-differentiated effects. The author successfully integrates epidemiological evidence with molecular insights and critically evaluates current toxicological benchmarks.
The manuscript is scientifically rigorous, clearly written, and of high relevance to environmental health, toxicology, nephrology, and public health policy. Figures and tables are informative, and the conclusions are well-supported by the literature.
Recommendation:
I recommend acceptance.
RESPONSE: Thank you for your recommendation.

Reviewer 3 Report
Comments and Suggestions for Authors
The paper discusses a major public health issue—cadmium exposure and its effects on kidney function—while also revealing a possibly novel molecular link involving 20-Hydroxyeicosatetraenoic acid, a physiologically active eicosanoid. While the research issue is topical and significant, the publication might benefit from greater clarity, mechanistic depth, and methodological transparency.
In abstract verify that the mechanical relationship between cadmium exposure and 20- Hydroxyeicosatetraenoic is clearly articulated (for example, is 20-Hydroxyeicosatetraenoic acid increased in response to cadmium?). Briefly describes the theory and significant findings. If the abstract lacks numerical results, include them to support important statements.
What is the short form of 20-Hydroxyeicosatetraenoic acid?
Emphasizes on cadmium's nephrotoxicity and the growing significance of lipid mediators such as 20- Hydroxyeicosatetraenoic.
Explain the physiological and pathological roles of cadmium in the kidney for example, regulating vascular tone and sodium processing.
More information is needed to understand why 20- Hydroxyeicosatetraenoic was suggested to mediate cadmium toxicity particularly.
Figures 2, 4, 5, and 6 are without references; if generated using the authors' own data, please provide methodology.
Comments on the Quality of English Language
Can be improved.
Author Response
Reviewer 3
Point 1:
The paper discusses a major public health issue—cadmium exposure and its effects on kidney function—while also revealing a possibly novel molecular link involving 20-Hydroxyeicosatetraenoic acid, a physiologically active eicosanoid. While the research issue is topical and significant, the publication might benefit from greater clarity, mechanistic depth, and methodological transparency.
RESPONSE:
Thank you for your comments and suggestions to improve a review. To better reflect the content of the present review, its objectives have been rewritten and its central working hypothesis has been depicted in a new Figure 1. The objective statements (lines 59-65) are quoted below.
“The principal objective of the present review is to provide a comprehensive knowledge on Cd exposure and its implications for kidney function and hypertension, with a specific focus on the mechanistic role of the eicosanoid 20-hydroxyeicosatetraenoic acid (20-HETE) and gender differentiated effects. It integrates epidemiological evidence with molecular insights and critically evaluates current toxicological benchmarks.”
Point 2:
In abstract verify that the mechanical relationship between cadmium exposure and 20- Hydroxyeicosatetraenoic is clearly articulated (for example, is 20-Hydroxyeicosatetraenoic acid increased in response to cadmium?). Briefly describes the theory and significant findings. If the abstract lacks numerical results, include them to support important statements.
RESPONSE: The purpose of abstract is to provide a succinct background and the scope of a review with 20-HETE is briefly introduced. It its present form, the abstract serves such purpose.
Point 3
What is the short form of 20-Hydroxyeicosatetraenoic acid?
RESPONSE: 20-HETE.
Point 4:
Emphasizes on cadmium's nephrotoxicity and the growing significance of lipid mediators such as 20- Hydroxyeicosatetraenoic.
RESPONSE:
Thank you for your suggestions. They will form a basis for future original research articles, examining the potential involvement of lipid mediators in the pathogenesis of Cd nephropathy.
Point 5.
Explain the physiological and pathological roles of cadmium in the kidney for example, regulating vascular tone and sodium processing.
RESPONSE: A new Figure 1 has been added to the Introduction to clarity the central hypothesis of the present review, involving salt and water balance by PT and DT.
Point 6.
More information is needed to understand why 20- Hydroxyeicosatetraenoic was suggested to mediate cadmium toxicity particularly.
RESPONSE: To the best of my knowledge, there was only one human study that investigated the potential role for 20-HETE in Cd effects on hypertension [113]. More research is required to entertain the reviewer suggestion.
[113] Boonprasert, K.; Vesey, D.A.; Gobe, G.C.; Ruenweerayut, R.; Johnson, D.W.; Na-Bangchang, K.; Satarug, S. Is renal tubular cadmium toxicity clinically relevant? Clin. Kidney J. 2018, 11, 681-687.
Point 7:
Figures 2, 4, 5, and 6 are without references; if generated using the authors' own data, please provide methodology.
RESPONSE: Figures 1, 2 and 4 are original. For the reminder Figures 3, 5-7, references to data sources have been added to the figure legends.
Point 8.
Comments on the Quality of English Language
Can be improved.
RESPONSE: I have carefully checked throughout a paper typo and grammatical errors. Necessary corrections have been undertaken. For clarity, some statements are rephrased.

Reviewer 4 Report
Comments and Suggestions for Authors
This is a relatively interesting review on a possible relationship between cadmium-derived hypertension and 20-hydroxyeisosatetraenoic acid. The postulate is adequately put forward and elucidated; it is also supported by the background material. Overall the manuscript is generally well written; however, several minor grammatical errors exists. Proofreading by someone extremely familiar with English grammar would be helpful.
My biggest concern is the figures, which need some attention. The author needs to try to place themself in a position of a potential reader who is not familiar with the field. Can they tell what is going on? Can they read everything? Can they see the caption at the same time they are looking at the figure so they can work out what is happening? When data is presented, the captions should have a reference!!! Also, permission to use the figure from a previously published work should be obtained; or if obtained, the permission should be indicated.
Figure 1 needs to have all the different components identified. How is one not familiar with this system supposed to know the glomerulus is depicted top middle and what its relationship to what is shown below is?
Figure 2 - do you have permission to reproduce? Caption should be entirely on page with figure.
Figure 3 - identify all parts and put caption on same page as figure. Text inside the nucleus is illegible.
Figure 4 - identify all parts. Some text is essentially illegible.
Figures 5 and 6 - do you have permission to reproduce?
Comments on the Quality of English Language
The English language is adequate; however, a careful review of the proper use of commas (both absent and extraneous) would be helpful. The errors in comma usage were most frustrating during my reading the manuscript. Also, the readability of the manuscript could be significantly improved if sentences that started indefinitely (e.g., "It can be...." and "There is...") could be eliminated.
Author Response
Reviewer 4
This is a relatively interesting review on a possible relationship between cadmium-derived hypertension and 20-hydroxyeisosatetraenoic acid. The postulate is adequately put forward and elucidated; it is also supported by the background material. Overall the manuscript is generally well written; however, several minor grammatical errors exists. Proofreading by someone extremely familiar with English grammar would be helpful.
My biggest concern is the figures, which need some attention. The author needs to try to place themself in a position of a potential reader who is not familiar with the field. Can they tell what is going on? Can they read everything? Can they see the caption at the same time they are looking at the figure so they can work out what is happening? When data is presented, the captions should have a reference!!! Also, permission to use the figure from a previously published work should be obtained; or if obtained, the permission should be indicated.
RESPONSE: Thank you for your advice and suggestions to improve Figures and their descriptions. Copyright issue is not involved because figures 1,2 and 4 are original. In the initial submission, original figures are indicated in the Acknowledgement of colleague who generated them. Such statement has been maintained (lines 544-545). The remainders figures 3, 5-7 are reproduced/modified from papers, published in Open Access Journals. References to these figures have now been provided.
Figure 1 needs to have all the different components identified. How is one not familiar with this system supposed to know the glomerulus is depicted top middle and what its relationship to what is shown below is?
RESPONSE: Figure 1 referred to is now named Figure 2. All components have been added. This Figure is original.
Figure 2 - do you have permission to reproduce? Caption should be entirely on page with figure.
RESPONSE: The referred Figure 2 becomes Figure 3. Reference to data source has been indicated (lines 243-244).
Figure 3 - identify all parts and put caption on same page as figure. Text inside the nucleus is illegible.
RESPONSE: A referred Figure 3 becomes Figure 4. This figure is original. The molecular targets of Cd and text inside the nucleus are described in caption (lines 288-292).
Figure 4 - identify all parts. Some text is essentially illegible.
RESPONSE: A referred Figure 4 becomes Figure 5. To improve legibility, the font size has been enlarged Reference to Figure source has been specified (lines 422-423).
Figures 5 and 6 - do you have permission to reproduce?
RESPONSE. Referred Figures 5 and 6 have been changed to Figures 6 and 7. Reference to the data sources are indicated on lines 447-448 and lines 453-454 for Figures 6 and 7, respectively.
Comments on the Quality of English Language
The English language is adequate; however, a careful review of the proper use of commas (both absent and extraneous) would be helpful. The errors in comma usage were most frustrating during my reading the manuscript. Also, the readability of the manuscript could be significantly improved if sentences that started indefinitely (e.g., "It can be...." and "There is...") could be eliminated.
RESPONSE: I have carefully checked for the errors to which the reviewer has referred. Necessary corrections have been undertaken.

Reviewer 5 Report
Comments and Suggestions for Authors
In the introductory part, the type of research paper should be mentioned, namely that it is a mini-review based on a hypothesis In the introduction, the real sources of human cadmium contamination should be mentioned In the conclusions, it should be mentioned that there are not enough studies to support the hypothesis

Author Response
Reviewer 5
Item 1. In the introductory part, the type of research paper should be mentioned, namely that it is a mini-review based on a hypothesis.
RESPONSE: Thank you for your comments and suggestions to improve a review. The objectives of the present review have been rewritten to better reflect that it is a comprehensive review (lines 59-65). To further improve it, a new Figure 1 is added to show the central working hypothesis regarding Cd-induced hypertension (lines 59-74). The objective statements are quoted below.
“The principal objective of the present review is to provide a comprehensive knowledge on Cd exposure and its implications for kidney function and hypertension, with a specific focus on the mechanistic role of the eicosanoid 20-hydroxyeicosatetraenoic acid (20-HETE) and gender-differentiated effects. It integrates epidemiological evidence with molecular insights and critically evaluates current toxicological benchmarks.”
Item 2. In the introduction, the real sources of human cadmium contamination should be mentioned.
RESPONSE: The anthropogenic, geologic, and other environmental causes of Cd contamination can be found in reference 2. This reference provides a good coverage on the global scale of Cd pollution.
[2] Hou, D.; Jia, X.; Wang, L.; McGrath, S.P.; Zhu, Y.G.; Hu, Q.; Zhao, F.J.; Bank, M.S.; O'Connor, D.; Nriagu, J. Global soil pollution by toxic metals threatens agriculture and human health. Science 2025, 388, 316–321.
Item 3. In the conclusions, it should be mentioned that there are not enough studies to support the hypothesis.
RESPONSE: Sentences quoted below are added to the Conclusion (lines 500-504).
“Evidence that 20-HETE contributes to blood pressure homeostasis comes from human gene mutation research, in vitro and animal studies. Studies into its role in gender differential effects of Cd in humans are limited. Further molecular mechanistic dissections in humans are warranted to pave the way to novel dietary interventions and therapies, especially for resistance hypertension.”

Reviewer 6 Report
Comments and Suggestions for Authors
The author’s purpose of the review paper entitled “ Hypertension in People Exposed to Environmental Cadmium: Roles for 20-Hydroxyeicosatetraenoic Acid in the Kidney” is timely and interesting also from related research fields.
Suggestions:
- At the end of the introduction, it is not clear what is the main message and relevant points of the paper that should be emphasize at this stage. What is timely and new? Specific objectives are not completely explicit.
- Also at the introduction or at section 2, a timeline figure of the historical contex of Cd use should be inserted and described in the body text and also with milestones for pathologies associated with Cd?
- Another figure that would be also pedadogical and to be discussed in the review would be a figure for the sources of Cd poisoning.
- At Figure 1 and figure 3, perhaps it should be better to include numbers, such as 1,2, 3 for each target namely: 1. Tubular cells; 2. Mitochondria; 3. Na/K ATPase…etc.
- Perhaps it should also be included in the review paper a small section about the Cd metalobiochemistry referring for instance to the competition with calcium and to calcium binding proteins thus affecting many processes associated with Ca2+ signaling, amomg others?
- Missing also a globall conclusion at conclusions, after the partial conclusions?
Author Response
Reviewer 6
The author’s purpose of the review paper entitled “Hypertension in People Exposed to Environmental Cadmium: Roles for 20-Hydroxyeicosatetraenoic Acid in the Kidney” is timely and interesting also from related research fields.
Suggestions:
Item 1. At the end of the introduction, it is not clear what is the main message and relevant points of the paper that should be emphasize at this stage. What is timely and new? Specific objectives are not completely explicit.
RESPONSE: Thank you for your comments and suggestions to improve a review. The objectives of the present review have been rewritten (lines 59-65). A new Figure 1 is added to provide the central working hypothesis regarding Cd-induced hypertension (lines 59-74). The objective statements are quoted below.
“The principal objective of the present review is to provide a comprehensive knowledge on Cd exposure and its implications for kidney function and hypertension, with a specific focus on the mechanistic role of the eicosanoid 20-hydroxyeicosatetraenoic acid (20-HETE) and gender-differentiated effects. It inte-grates epidemiological evidence with molecular insights and critically evaluates current toxicological benchmarks.”
Item 2. Also at the introduction or at section 2, a timeline figure of the historical context of Cd use should be inserted and described in the body text and also with milestones for pathologies associated with Cd?
RESPONSE: The history topic of Cd has been covered by a review by others.
Nordberg M, Nordberg GF. Metallothionein and Cadmium Toxicology-Historical Review and Commentary. Biomolecules. 2022 Feb 24;12(3):360.
Item 3. Another figure that would be also pedagogical and to be discussed in the review would be a figure for the sources of Cd poisoning.
RESPONSE: Food as a source of Cd exposure is now indicated in a new Figure 1 together with transport proteins for Fe, Zn, Ca that are involved in the assimilation of Cd.
Item 4. At Figure 1 and figure 3, perhaps it should be better to include numbers, such as 1,2, 3 for each target namely: 1. Tubular cells; 2. Mitochondria; 3. Na/K ATPase…etc.
RESPONSE: The referred Figures 1 and 3 have become Figure 2 and 4 in a revised version. I appreciate the reviewer’s suggestions. However, as it is briefly discussed in the third paragraph of the Introduction (lines 49-58), Cd targets are numerous. I have now provided in the text the essential targets in the caption of Figures 2 and 4.
Item 5. Perhaps it should also be included in the review paper a small section about the Cd metalobiochemistry referring for instance to the competition with calcium and to calcium binding proteins thus affecting many processes associated with Ca2+ signaling, among others?
RESPONSE: As implied by a new Figure 1, Cd can compete with Ca, Zn and Fe. Given that not only Ca, but also Zn and Fe that can affect Cd toxic outcomes, it would be misleading to single out Ca.
Item 6. Missing also a global conclusion at conclusions, after the partial conclusions?
RESPONSE: A global conclusion has now been provided in the last paragraph of the Conclusion, quoted below.
“Current environmental Cd exposure has now reached toxic levels in a significant proportion of the population. No consensus on Cd exposure limits exists nor its permissible levels in food and water. The Australian and New Zealand Standard for Cd in drinking water is 2 µg/L, lower than 5 µg/L of the WHO drinking water guidelines. Evidence suggests also that a decrease in eGFR due to Cd-induced nephron destruction is irreversible. New Cd exposure guidelines should be established. An effective chelation therapy to remove Cd from the kidneys does not exist. Essential preventive measures should include avoiding smoking and foods containing high Cd levels, keeping an optimal body weight, and minimizing Cd assimilation and kidney burden of Cd by maintaining body content of calcium, zinc, and iron.”

Round 2
Reviewer 3 Report
Comments and Suggestions for Authors
Authors responses are ok.
My only comment is on reference 122, the data taken from reference 122 like table 3 and figures 6 and 7 were not published in any journal, present on Research Square. It may create copyright problem.
Author Response
Reviewer 3
Authors responses are ok.
My only comment is on reference 122, the data taken from reference 122 like table 3 and figures 6 and 7 were not published in any journal, present on Research Square. It may create copyright problem.
Response: Reference 122 is an Open-Access research article, distributed under the terms and conditions of the Creative Commons Attribution License (CC BY). Permission/copyright are not required to reproduce Figures and Tables.
Reviewer 4 Report
Comments and Suggestions for Authors
Issues have been addressed.
Author Response
Reviewer 4
Issues have been addressed.
Response: Thank you for your evaluation of my work.
Reviewer 6 Report
Comments and Suggestions for Authors
The authors were very positive, although not agreeing with all the suggestions.
Author Response
Reviewer 6
The authors were very positive, although not agreeing with all the suggestions.
Response: Thank you for your evaluation of my work. I truly appreciate your comments and suggestions.